# Species-, organ- and cell-type-dependent expression of SPARCL1 in human and mouse tissues

Anika Klingler[1], Daniela Regensburger[1], Clara Tenkerian[1], Nathalie Britzen-Laurent[1], Arndt Hartmann[2], Michael Stürzl[1], Elisabeth Naschberger[1]*

1 Division of Molecular and Experimental Surgery, Department of Surgery, University Medical Center Erlangen, Friedrich-Alexander University of Erlangen-Nuremberg, Translational Research Center, Erlangen, Germany, 2 Institute of Pathology, University Medical Center Erlangen, Friedrich-Alexander University of Erlangen-Nuremberg, Erlangen, Germany

* Elisabeth.naschberger@uk-erlangen.de

**Data Availability Statement:** All relevant data are within the paper and its Supporting Information files.

**Funding:** This work was supported by grants of the Interdisciplinary Center for Clinical Research (IZKF,

## Abstract

SPARCL1 is a matricellular protein with anti-adhesive, anti-proliferative and anti-tumorigenic functions and is frequently downregulated in tumors such as colorectal carcinoma or non-small cell lung cancer. Studies have identified SPARCL1 as an angiocrine tumor suppressor secreted by tumor vessel endothelial cells, thereby exerting inhibitory activity on angiogenesis and tumor growth, in colorectal carcinoma. It is unknown whether SPARCL1 may exert these homeostatic functions in all organs and in other species. Therefore, SPARCL1 expression was comparatively analysed between humans and mice in a systematic manner. Murine Sparcl1 (mSparcl1) is most strongly expressed in the lung; expressed at an intermediate level in most organs, including the large intestine; and absent in the liver. In human tissues, SPARCL1 (hSPARCL1) was detected in all organs, with the strongest expression in the stomach, large intestine and lung, mostly consistent with the murine expression pattern. A striking difference between human and murine tissues was the absence of mSparcl1 expression in murine livers, while human livers showed moderate expression. Furthermore, mSparcl1 was predominantly associated with mural cells, whereas hSPARCL1 was detected in both mural and endothelial cells. Human SPARCL1 expression was downregulated in different carcinomas, including lung and colon cancers. In conclusion, this study revealed species-, organ- and cell-type-dependent expression of SPARCL1, suggesting that its function may not be similar between humans and mice.

## Introduction

*Secreted protein acidic and rich in cysteines like 1* (SPARCL1, syn.: Sc1, hevin, MAST9) is a matricellular protein belonging to the SPARC protein family, and SPARC/osteonectin is its closest and most prominent family member [1, 2]. Matricellular proteins are secreted proteins present in the extracellular matrix [3]. They have de-adhesive activity, in contrast to the adhesive extracellular matrix proteins fibronectin, vitronectin, and collagen [3]. Human SPARCL1

D28 to EN/MS, D34 to MS, J73 to CT, stipendia to AK) of the University Medical Center Erlangen, the German Research Foundation (DFG: FOR 2438, sub-project 2 to EN/MS; TRR 241, sub-project A06 to MS/NBL; KFO 257, sub-project 4 to MS; SFB 796, sub-project B9 to MS), the W. Lutz Stiftung to MS and the Forschungsstiftung Medizin am Universitätsklinikum Erlangen to MS.

**Competing interests:** The authors have declared that no competing interests exist.

consists of an N-terminal secretion signal peptide causing an internal follistatin-like domain (FLD), a C-terminal extracellular calcium-binding domain and a highly acidic domain positioned between the signal peptide sequence and the FLD that is 411 amino acids long in SPARCL1 but only 51 amino acids in SPARC [1].

Only limited and partly conflicting information is available on the expression pattern of SPARCL1 in humans and mice. In agreement with the initial isolation of human SPARCL1 (hSPARCL1) from high endothelial venules, subsequent publications showed that hSPARCL1 expression in different tumors, including colorectal carcinoma (CRC), is highly associated with blood vessel endothelial cells [1, 4]. In culture, hSPARCL1 expression is induced in quiescent endothelial cells (ECs) but absent in actively proliferating ECs [5]. However, with the exception of pancreatic carcinoma cells, SPARCL1 was not found to be expressed in many other cell types investigated [4]. Initially, conflicting results have been reported on hSPARCL1 expression in tumor cells in CRC tissues [6, 7]. This was paralleled by conflicting results on the association of hSPARCL1 expression with the prognosis of CRC patients [6, 7]. These contradictory findings may have originated from nonspecific staining signals. The initial study on hSPARCL1 expression in CRC detected a strong association with tumor vessel endothelial cells (TECs) using in situ hybridization (ISH) [4]. These findings were confirmed by our group using ISH and at the protein level with immunohistochemistry (IHC) [5]. In human CRC tissues, hSPARCL1 was identified to be preferentially expressed by endothelial (EC) and mural cells in CRC but not by the tumor cells themselves [5]. In this study, staining controls such as isotype controls were included, and independent methods (IHC, ISH) were used. Therefore, the previously described epithelial/tumor cell signal in the colon and CRC was demonstrated to be nonspecific. Accordingly, the previously reported associations of hSPARCL1 expression with a specific CRC patient prognosis now require validation. Moreover, hSPARCL1 expression was found to be retained in favourable Th1 tumor microenvironments (TME) in CRC patients similar to the normal colon but to be preferentially lost in aggressive TMEs. Notably, *in vitro* hSPARCL1 expression was then identified to be induced by endothelial cell quiescence and was further stabilized by the addition of the Th1 cytokines interferon (IFN)-γ and/or interleukin (IL)-2, which are present in favourable Th1-TMEs of CRC patients [5]. Human SPARCL1 expression is also commonly downregulated in other cancer tissues, including CRC, metastatic prostate adenocarcinoma [8, 9], non-small cell lung cancer [10], metastases of pancreatic cancer [11], gastric cancer [12, 13], breast cancer [14], and hilar cholangiocarcinoma [15].

In contrast, mouse Sparcl1 (mSparcl1) expression has been examined in only a few studies, mostly at the RNA level. Mouse Sparcl1 mRNA was found to be highly expressed in the brain, at moderate levels in the lung, heart and adrenal gland and at low levels in the kidney, eye, liver, submandibular gland and testis [16]. Expression was mainly localized in the media and adventitia layers of medium and larger vessels as well as in the cardiac muscle and the bronchial tube system of the lung [16]. A limited number of murine organs, such as the brain, eye, heart and lung, were also analysed by western blot and found to express mSparcl1 at high or intermediate levels [17]. The presence of mSparcl1 protein at the single-cell level in different organs has not yet been investigated.

Notably, full-length mSparcl1 migrates at approximately 120–130 kDa in western blots. Additional mSparcl1-specific bands between 40–55 kDa in size with an unknown function have been described [17, 18]. It has been shown that these proteins may be generated by cleavage with ADAMTS4 [17] or matrix metalloproteinase-3 [19] preferentially in the brain, resulting in a fragment with a similar molecular weight to that of SPARC.

Regarding SPARCL1 function, it was reported that hSPARCL1 and mSparcl1 inhibit endothelial cell attachment and spreading on fibronectin [20] and adhesion of murine dermal

fibroblasts during wound closure, respectively [21]. Mice with a double knockout of mSparcl1 and mSparc showed increased vessel density during a foreign body response reaction, indicating anti-angiogenic activity of one or both of the proteins [22]. Interestingly, mSparcl1 induces the formation of synapses between cultured rat ganglion cells, whereas mSparc antagonizes this effect, indicating that the two proteins can have opposite activities [3, 23]. However, at the tissue level, mSparc and mSparcl1 are generally not coexpressed. For example, mSparc is highly expressed in the placenta and testis and at low levels in the brain, whereas mSparcl1 is highly expressed in the brain, colon, heart and lungs and at low levels in the placenta and testis [1]. No cellular receptor has been identified for human or mouse SPARCL1 as yet. It has been shown that hSPARCL1 inhibits cell proliferation by prolonging the G1 phase in HeLa cells [24]. Recent studies revealed that hSPARCL1 in CRC exerts both anti-angiogenic [5] and anti-tumorigenic [6] functions. In this context, hSPARCL1 actively maintains vessel homeostasis in the CRCs of patients with favourable prognosis by inducing EC quiescence and stabilization of mature vessels by acting on both endothelial and mural cells [5].

Overall, human SPARCL1 can be considered a vascular-derived and therefore angiocrine tumor suppressor. Accordingly, its presence and potentially different cell type-associated expression patterns in the tissues of humans and mice are of great interest for future mechanistic *in vivo* studies using transgenic mouse models. The aim of the present study was to comparatively investigate SPARCL1 expression between humans and mice and to systematically analyse it at the RNA and protein levels with different techniques to identify overlapping and differential SPARCL1 expression patterns between the different species.

## Materials and methods

### Tissues from human patients and mice

Human tissues were retrieved after completion of routine diagnostics as formalin-fixed, paraffin-embedded blocks from the Institute of Pathology, Friedrich-Alexander University Erlangen-Nuremberg. Murine tissues were harvested from Sparcl1/Sc1 knockout C57BL/6 mice [5] (n = 3) and wild-type animals (n = 3) and were either immediately processed to obtain formalin-fixed, paraffin-embedded tissue blocks or snap-frozen with liquid nitrogen for later RNA isolation. The procedures were approved by the local ethics committee (human tissues) and licensed by the local government (mouse tissues).

### Protein extraction and western blot analysis

Protein was extracted from fresh murine tissues by homogenization in RIPA buffer using a TissueLyser II (Qiagen, Hilden, Germany) at 25 Hz for 2 min, 30 min incubation on ice and 20 min of centrifugation at 4°C and 18,000 *xg*. The obtained supernatant was stored at -20°C until analysis. Protein concentrations in the tissue extracts were determined using the DC protein assay (Bio-Rad, Munich, Germany). The samples were boiled in Laemmli buffer for 5 min, size-separated by 10% SDS-PAGE, and electrophoretically transferred to a PVDF membrane (Carl Roth, Karlsruhe, Germany) at 250 mA for 2 h. After blocking in PBS-0.1% Tween containing 5% skim milk overnight at 4°C, the blots were incubated for 1 h at room temperature (RT) with the following primary antibodies: polyclonal goat anti-mouse Sparcl1 (R&D Systems, Minneapolis, USA, 1:500) and polyclonal rabbit anti-mouse GAPDH (Merck, Darmstadt, Germany, cat no. ABS16, 1:10,000). Detection of the primary antibodies was performed using rabbit anti-goat and goat anti-rabbit immunoglobulin G (IgG) coupled to horseradish peroxidase (HRP) (DakoCytomation, Santa Clara, USA, 1:5,000) for 45 min at RT. All antibodies were diluted in PBS-0.1% Tween/2.5% skim milk. The HRP enzyme reaction was

developed using enhanced chemiluminescence (ECL) prime reagents (GE Life Sciences, Chalfont St Giles, Great Britain) and recorded using the Amersham Imager 600 (GE Life Sciences).

## RNA isolation and quantitative reverse transcriptase-polymerase chain reaction (qRT-PCR)

RNA from murine tissues was isolated using the RNeasy Mini Kit (Qiagen). DNA digestion (Ambion, Carlsbad, USA) and glycogen purification (Thermo Scientific, Waltham, USA) were performed. RNA quantity was measured using a Nanodrop 2000c (Thermo Scientific). Primers and probes for qRT-PCR were designed using Primer-BLAST (NIH). The primers and probes were purchased from Eurogentec (Lüttich, Belgium). The RNA primer/probe sequences were as follows: mSparcl1: forward `CCTCTCCGCAGATCTAGCCA`, probe `TCCTCCTGTGCGCCTTGGGA`, reverse `CTTCCGGTGTCACCAGTGTT`; mGAPDH: forward `ACTGAGCAAGAGAGGCCCTA`, probe `TCCCAACTCGGCCCCCAACA`, reverse `TATGGGGGTCTGGGATGGAA`. qRT-PCR was performed as described previously using the SuperScript III Platinum One-Step Quantitative RT-PCR System with ROX (Life Technologies) [5]. In brief, a reaction (10 µl total volume) consisted of 0.2 µL RT/Taq-mix, 50 nM ROX Reference dye (both Life Technologies), 10 ng total RNA, 3.28 µl water, 500 nM forward/reverse primer and 250 nM probe. The reactions were assayed in triplicate in 96-well qPCR plates (Agilent Technologies, Santa Clara, USA) using an MX3005P qPCR system (Agilent Technologies) with Versant kPCR software (Siemens Healthcare Diagnostics, Erlangen, Germany). For the quantification of RNA, the absence of potential residual DNA was analysed by DNA-specific primers for the progestogen-associated endometrial protein (PAEP) gene: forward `CACAGAATGGACGCCATGAC`, probe `AAGCCCTCAGCCCTGCTCTCCATC`, reverse `AAACCAGAGAGGCCACCCTAA`; the results were negative in all cases. The fluorescence threshold (ROX dRn) was set to 0.02 for all samples. All samples were normalized using murine GAPDH (mean of triplicates) as a reference gene. The ΔCT method was used to calculate the respective relative expression values.

## Permanent immunohistochemistry (IHC)

Tissue blocks embedded in paraffin were cut into 4 µm tissue sections and transferred to slides (Superfrost Plus, Thermo Scientific). The tissue was deparaffinized with xylene two times for 15 min and rehydrated using decreasing concentrations of ethanol (100%, 96%, 85%, 70%) for 2 min each. Antigen retrieval was performed for both murine and human tissues using Target Retrieval Solution pH 9.0 (DakoCytomation) at 95°C for 20 min in a water bath followed by 20 min cooling at RT. The slides were washed in Tris buffered saline (TBS), pH 7.6, two times for 5 min. Blocking of the endogenous peroxidase was performed using 7.5% $H_2O_2$ in 70% ethanol for 10 min followed by three washes with A.d. and twice for 5 min of TBS. Nonspecific antigens were blocked using 2.5% normal horse serum for 20 min. Next, the sections were incubated with one of the following primary antibodies: polyclonal goat anti-mouse Sparcl1 (15 min, 1:1,000, cat. no. AF2836, R&D Systems) or polyclonal goat anti-human SPARCL1 (1 h, 1:900, cat. no. AF2728, R&D Systems). Normal goat IgG (in corresponding concentrations, cat. no. AB-108-C, R&D Systems) was used for isotype controls. All antibodies used were diluted in background reducing antibody diluent (DakoCytomation). The slides were washed 5 min two times in TBS and incubated for 30 min with ImmPRESS HRP Reagent (Vector Laboratories, Burlingame, USA). After washing the slides for 5 min two times in TBS, the substrate Nova Red (Vector Laboratories) was added for 15 min. Next, the slides were washed twice for 2.5 min in A.d., counterstained with Gill-III haematoxylin (Merck), dehydrated and mounted

with VectaMount Permanent Mounting Medium (Vector Laboratories). The tissues were analysed by a DM6000 B microscope (Leica, Wetzlar, Germany) and LAS-X software (Leica).

## Fluorescent immunohistochemistry (F-IHC)

Cutting, deparaffinizing, rehydration and antigen retrieval were performed as described above. Slides were washed in 0.45 μm filtered TBS, pH 7.6, two times for 5 min. The slides were blocked in 10% donkey normal serum (DNS; Dianova, Hamburg, Germany) diluted in filtered TBS for 10 min. Next, the tissues were incubated with the following primary antibodies in different combinations diluted in 5% DNS for 1 h at RT: polyclonal goat anti-human SPARCL1 (1:50, cat. no. AF2728, R&D Systems), polyclonal rabbit anti-human CD31 (1:50, cat. no. RB-10333-P, Thermo Scientific), polyclonal goat anti-murine Sparcl1 (1:50, cat. no. AF2836, R&D Systems) and monoclonal rat anti-mouse CD31 (1:100, clone SZ31, Dianova). For isotype controls, the following antibodies were used: normal goat IgG (in corresponding concentrations, cat. no. AB-108-C, R&D Systems), mouse IgG1 isotype control (in corresponding concentrations, cat. no. MAB002, R&D Systems) and rat IgG2a (in corresponding concentrations, cat. no. MAB006, R&D Systems). The slides were washed twice for 5 min in filtered TBS and incubated with the following secondary antibodies in different combinations diluted in 5% DNS for 45 min at RT: Alexa Fluor 488 donkey anti-goat IgG (1:500, cat. no. A11055, Life Technologies), Alexa Fluor 546 donkey anti-rabbit IgG (1:500, cat. no. A10040, Life Technologies) and Alexa Fluor 555 donkey anti-rat IgG (1:500, cat. no. 6430–32, Southern Biotech, Birmingham, USA). The sections were washed twice for 5 min in filtered TBS and incubated with DRAQ5 (1:800, cat. no. 4084L, Cell Signaling, Danvers, USA) diluted in A.d. for 10 min. Next, slides were washed twice for 5 min in TBS and mounted with Fluorescence Mounting Medium (DakoCytomation). Confocal microscopy was performed on a Leica TCS SPE (Leica) with LAS-X software (Leica).

## Results

### Murine Sparcl1 expression is organ dependent

Sparcl1 expression in murine tissues was first examined at both the RNA and protein levels by qRT-PCR and western blot. At the mRNA level, organs from at least 3 independent wild-type C57BL/6NRj mice and corresponding Sparcl1 knockout animals were investigated, and the highest mSparcl1 expression levels were identified in the lung, followed by the stomach, spleen, large intestine (coecum/colon), and heart (Fig 1A, left panel: normalized Ct values; right panel: relative fold change). Murine Sparcl1 was only weakly expressed in the oesophagus, small intestine and kidney compared to the other organs. In the murine liver, mSparcl1 expression was absent (Fig 1A, +/+). In the knockout animals used as a control, all organs were negative (Fig 1A, left panel, KO). Western blot analysis of the same organs from 3 independent wild-type animals using a specific anti-murine Sparcl1 antibody revealed mSparcl1 to be expressed in most of the organs, and the results were relatively consistent compared to the RNA assay, namely, strong expression in the stomach, colon, and lung (Fig 1B, Sc1 +/+). Intermediate levels were detected in the coecum, whereas weak levels were detected in the oesophagus, heart and spleen. Similar to the RNA, mSparcl1 expression was absent in the liver (Fig 1B, liver). The low levels of RNA expression in the small intestine and kidney did not result in mSparcl1 expression detected at the protein level (Fig 1B). Notably, the large intestine with samples of the colon and coecum was the only organ clearly showing the previously described mSparcl1--specific band migrating at the lower size of 55 kDa [18] (Fig 1B, mSparcl1 fragment). As a control, lysates from the corresponding organs of mSparcl1-knockout animals were used and were negative in all cases (Fig 1B, Sc1-/-).

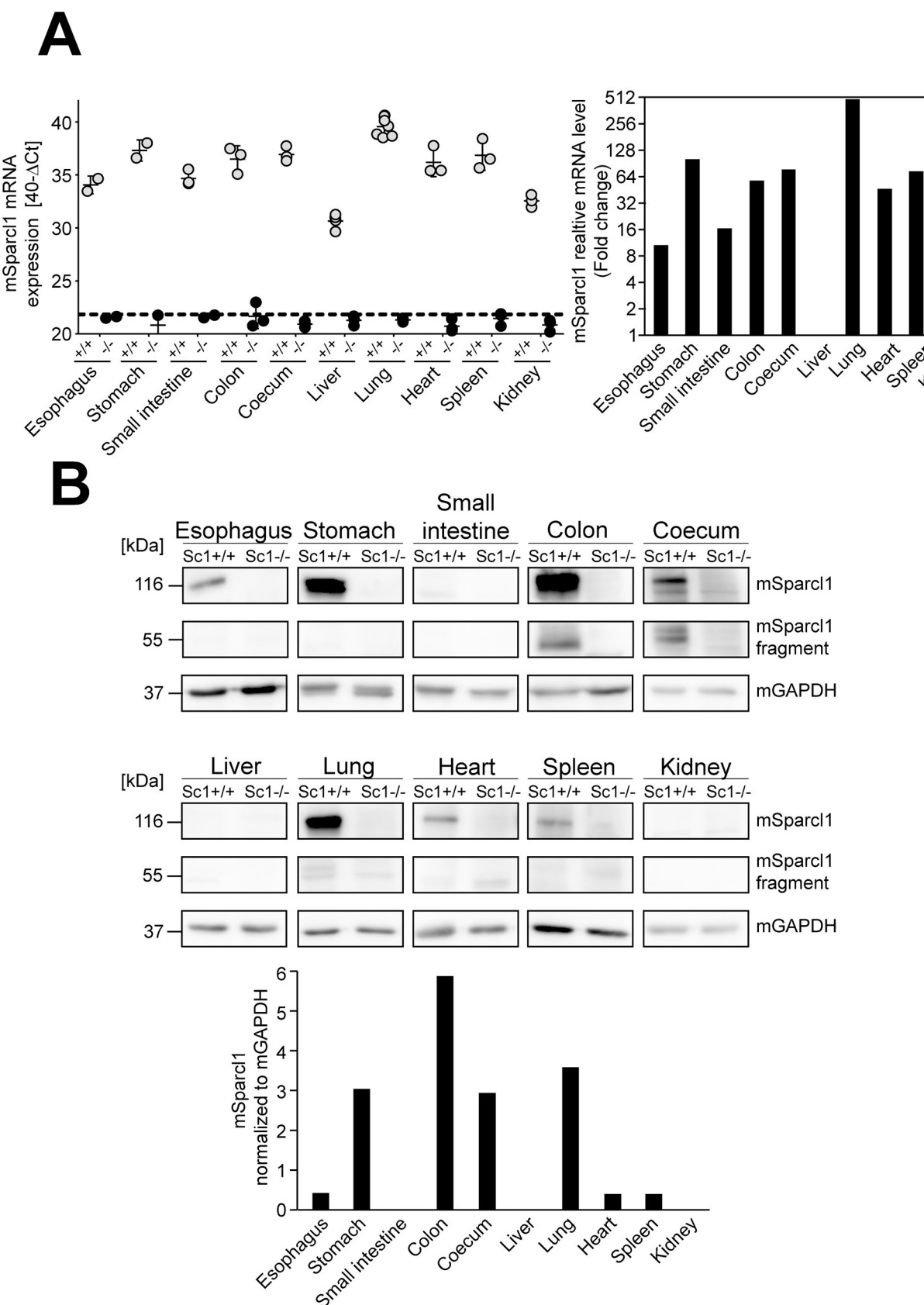

**Fig 1. Murine Sparcl1 is expressed at different levels in murine organs. (A)** Quantification of mSparcl1 mRNA expression by RT-qPCR in different organs of wild-type (Sc1 +/+, grey dots) and SPARCL1 knockout (Sc1 -/-, black dots) animals. The relative mRNA expression levels are indicated as 40-ΔCt values for each animal (left panel) and for wild-type animals only as the mean fold change calculated relative to the liver (right panel). **(B)** Western blot analysis of mSparcl1 protein expression in different murine organs of Sparcl1 wild-type (Sc1 +/+, n = 3) and knockout (Sc1 -/-, n = 3) mice. The results of one representative animal for each genotype are depicted. Murine GAPDH was detected as a loading control. The mSparcl1 signal (large band) of the depicted western blot was quantified and is given as the relative mSparcl1 signal after normalization to mGAPDH.

## Murine Sparcl1 is preferentially expressed by mural cells

F-IHC and permanent IHC were conducted to elucidate the expression pattern of mSparcl1 at the single-cell level. In accordance with the western blot results, mSparcl1 expression was detected strongly in the lung; at intermediate levels in the stomach, large intestine (coecum/colon), heart and urinary bladder; and weakly in the oesophagus, small intestine, spleen and kidney (Fig 2; S1 Fig; Sc1 +/+, green signal). Co-staining of mSparcl1 with CD31 was performed to evaluate potential vessel-associated expression of mSparcl1 as suggested by our previous data from the colon. This analysis revealed that mSparcl1 was associated with vessels in all organs except the lung and was absent in the liver. The vessel-associated mural cells of the oesophagus, stomach, large intestine (coecum/colon), lung, heart, spleen, kidney and urinary bladder were predominantly mSparcl1-positive (Fig 2; S1 Fig; Sc1 +/+). Endothelial cells expressed mSparcl1 only in the oesophagus, small intestine and large intestine (Fig 2; Sc1 +/+, yellow signal). Moreover, mSparcl1 was expressed by the stromal cells of the stomach, small intestine, large intestine and urinary bladder, by muscle cells of the heart, and in the lung around the bronchial tubes and in the alveolar tissue (Fig 2; S1 Fig; Sc1 +/+). Control isotype antibody staining was performed for all the organs and was negative (Fig 2; S1 Fig; Sc1 +/+ isotype). Furthermore, mSparcl1 knockout mice were stained for CD31 and mSparcl1 as an additional control. Staining was also negative for mSparcl1 in all cases (Fig 2; S1 Fig; Sc1 -/).

In addition, permanent IHC was conducted as a control. Consistently, immunofluorescence showed that the expression of mSparcl1 was strongest in the lung with specific signals in the alveolar tissue and around the bronchial tubes (S2 Fig; Sc1 +/+, brown signal). Intermediate mSparcl1 expression levels were revealed in the large intestine (coecum/colon), heart and kidney. The small intestine and spleen showed low mSparcl1 expression levels. In these organs, mSparcl1 expression was located at the vessels and in other stromal cells, likely smooth muscle cells. In contrast to the immunofluorescent staining results, weak mSparcl1-specific staining was observed in hepatocytes of the liver. Isotype staining (S2 Fig; Sc1 +/+ isotype) and staining of mSparcl1 knockout mice (S2 Fig; Sc1 -/-) were used as controls. These results revealed non-specific staining of the epithelium in the small intestine and large intestine, throughout the tissue of the spleen and in the tubule cells of the kidney (S2 Fig; asterisks).

## Human SPARCL1 is preferentially expressed by endothelial and mural cells in an organ-dependent manner

In a next step, different tissues of human patients were stained for hSPARCL1 by F-IHC and permanent IHC. These staining assays revealed hSPARCL1 to be expressed in all organs at different levels (Fig 3, healthy, green signal). The strongest expression of hSPARCL1 was detected in the stomach, large intestine, and lung, followed by an intermediate expression level in the oesophagus, small intestine and liver (Fig 3). Co-staining for hSPARCL1 and CD31 was conducted to determine vessel-associated hSPARCL1 expression. Human SPARCL1 expression was associated with the endothelial cells of the small intestine, large intestine, and lung (Fig 3, healthy, yellow signal). Human SPARCL1 was also detected in the mural cells of all organs examined. Furthermore, other stromal cells, likely smooth muscle cells of the oesophagus,

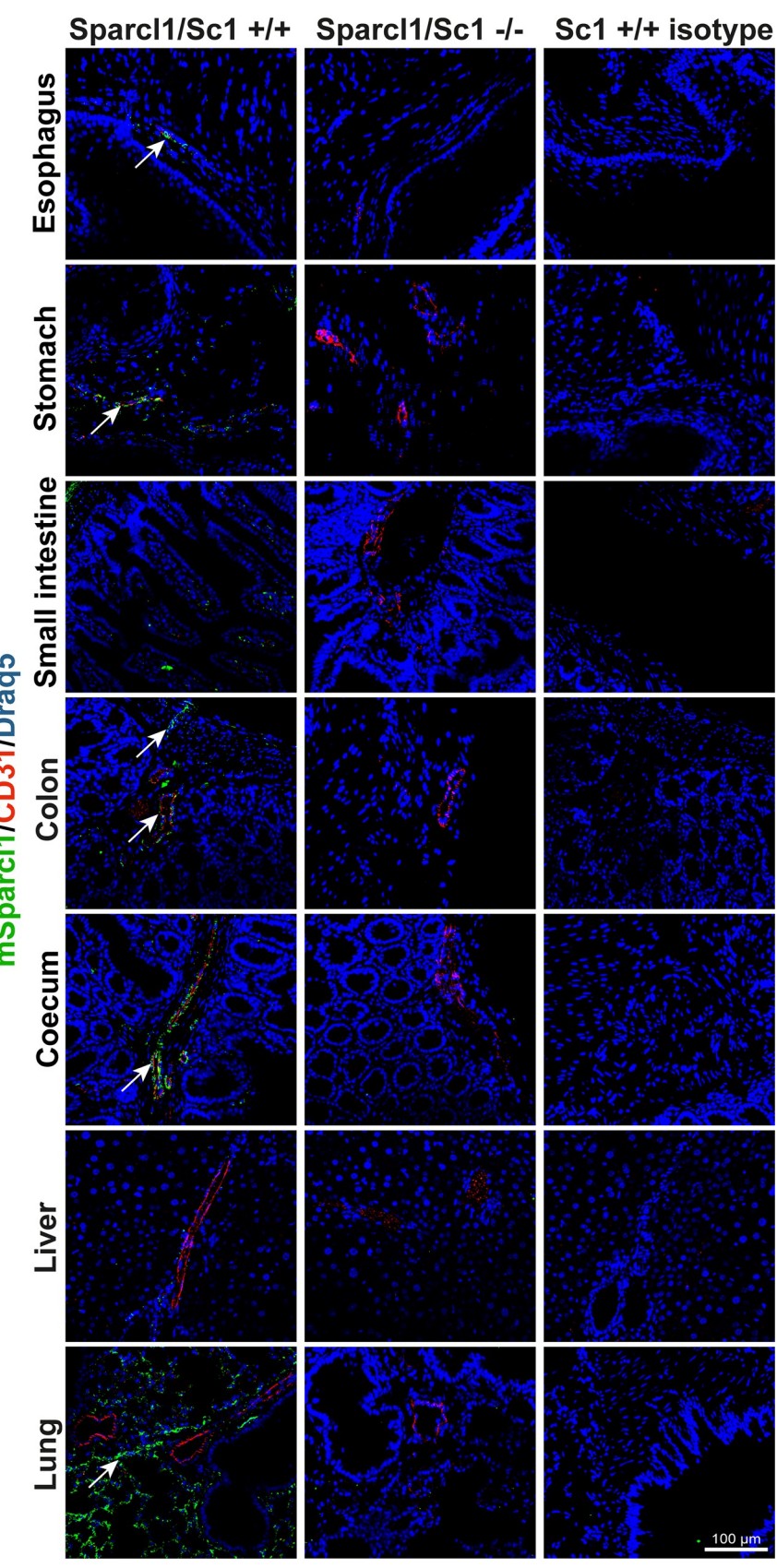

**Fig 2. Murine Sparcl1 is expressed in all murine organs examined except the liver.** Murine Sparcl1 expression (Sc1 +/+, green, arrows) was determined by immunofluorescence in various organs of Sparcl1 wild-type mice (n = 3). Isotype antibody (Sc1 +/+ isotype) and knockout mouse (Sc1-/-, n = 3) staining were used as controls. All tissues were counterstained using DRAQ5 (blue). Scale bar = 100 μm.

stomach, small intestine and large intestine, expressed hSPARCL1 occasionally. The pancreas showed hSPARCL1 expression in the islet cells and the lung around the bronchial tubes and in the alveolar tissue (Figs 3 and 4). Notably, as a control, isotype antibody control staining was performed for all the organs and was negative in all cases (Fig 3; isotype).

In addition, permanent IHC was conducted as a control. This analysis also revealed hSPARCL1 to be vastly expressed in all organs examined, consistent with the results of F-IHC. Here, again, hSPARCL1 expression was shown to be preferentially associated with vessels (S3 Fig; healthy). Furthermore, hSPARCL1-specific staining was located in the epithelium of the oesophagus, smooth muscle cells of stomach, small and large intestine, islet cells of the pancreas, hepatocytes of the liver and alveolar tissue of the lung as well as around the bronchial tubes. Isotype antibody control staining (S3 Fig; isotype) showed that the hSPARCL1 staining of the glandular cells of the stomach was nonspecific.

### Human SPARCL1 is downregulated in malignant diseases

To estimate potential differences in human SPARCL1 expression between healthy tissues and malignant diseases, corresponding malignant tissue sections of the abovementioned healthy organs were analysed. Both permanent IHC (S3 Fig; brown signal) and F-IHC showed consistently that hSPARCL1 expression is reduced in the corresponding malignomas of stomach, small and large intestine, and lung (Fig 3; neoplastic; Table 1) when compared with the healthy tissue. Additionally, for malignomas, co-staining for hSPARCL1 and CD31 was conducted to estimate the association with vessels. Human SPARCL1 was expressed in the endothelial cells of malignant tissues from the oesophagus, stomach, and lung (Fig 3; neoplastic, yellow signal). Permanent IHC consistently identified hSPARCL1 as expressed in a similar pattern to that detected by immunofluorescent staining (S3 Fig; neoplastic). Furthermore, in malignomas, hSPARCL1-specific staining was located in the stromal cells of the large intestine, hepatocytes of the liver and alveolar tissue of the lung.

### Liver and pancreas show differential SPARCL1 expression between mouse and human

Comparison of healthy murine and human tissues revealed that SPARCL1 is differentially expressed. The most striking difference that was observed was in the overall expression levels in the pancreas and liver (Fig 4). The murine pancreas showed only weak mSparcl1 expression at the protein level by IHC, whereas the healthy human pancreas expressed hSPARCL1 strongly (Figs 3 and 4). Likewise, the murine liver did not express mSparcl1 at all, in contrast to the human liver, which was found to have intermediate SPARCL1 expression as detected by IHC (Figs 1, 3 and 4). The human small intestine showed slightly higher hSPARCL1 expression than the corresponding murine organ. All other organs examined, such as the stomach, large intestine and lung, exhibited similar SPARCL1 expression levels and patterns (Fig 4).

Moreover, fluorescent co-staining of SPARCL1 and CD31, an endothelial cell marker, revealed that SPARCL1 is expressed in a vessel-associated manner according to our previous data. Human SPARCL1 was found to be colocalized with CD31 and hence expressed in endothelial cells in organs of the gastrointestinal tract, namely, the stomach, small intestine and large intestine (Fig 4; Table 2). However, in mice, vascular-associated mSparcl1 was mostly

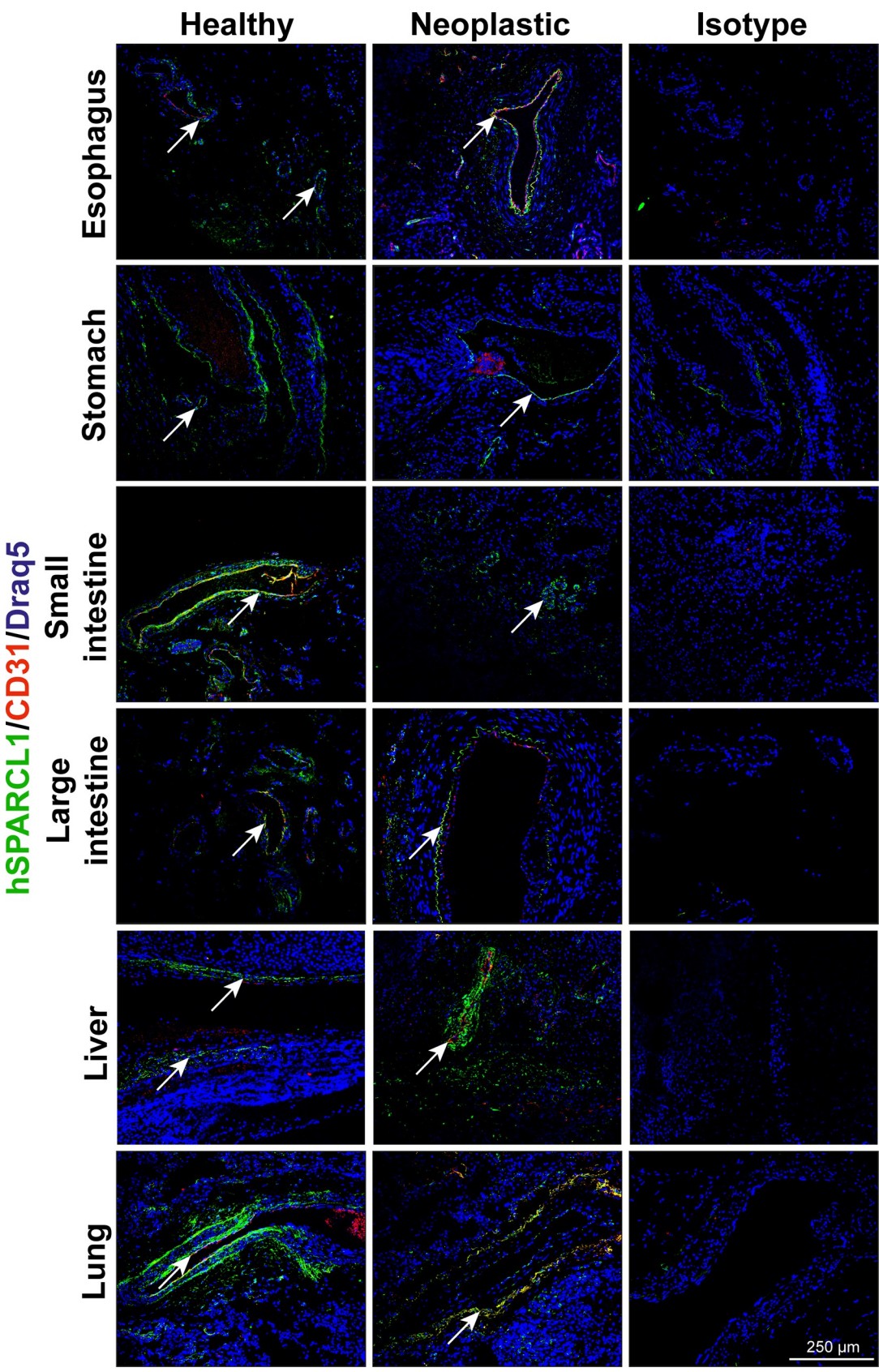

**Fig 3. Human SPARCL1 expression is highly detectable in all human organs examined and downregulated in their respective neoplasms.** Human SPARCL1 expression (green, arrows) was determined by immunofluorescence in various healthy and neoplastic organs of human patients (healthy, neoplastic, n = 3 for all organs except oesophagus, small intestine and lung tumor with n = 2). Isotype antibody staining of consecutive sections was used as a negative control (isotype). All tissues were counterstained using DRAQ5 (blue). Scale bar = 250 μm.

expressed by vessel-associated mural cells (Fig 4; Table 2). In contrast, the human tissues showed a different SPARCL1 expression pattern regarding vascular cell association. Here, all organs, except for the liver, showed SPARCL1 expression in both endothelial and mural cells (Fig 4, right; Table 2).

## Discussion

In this study, a species-, organ- and cell-type-dependent differential expression pattern of SPARCL1 between humans and mice was identified. SPARCL1 is an anti-angiogenic, anti-proliferative and anti-adhesive protein [5, 20, 21, 24]. Previous analyses of hSPARCL1 expression in human tissues showed a preferential downregulation of expression in neoplastic tissues [5, 8–15]. This is also reflected by recent studies examining the effects of hSPARCL1 on tumorigenesis and metastasis formation [25–30]. However, there is still little information on the general expression patterns of SPARCL1 in human and murine tissues. Here, we analysed the expression of SPARCL1 in various tissues of mice and humans and showed that hSPARCL1 and mSparcl1 expression patterns are different in a species- and cell type-dependent manner. This was accomplished with the help of F-IHC, permanent IHC, western blot analysis and qRT-PCR to investigate SPARCL1 expression at both the protein and mRNA levels. Notably, a comparison of previous studies indicated that in some instances, cross-reactivity of different antibodies might be the reason for differential staining results. To avoid this in our study, careful staining controls were applied. First, in all cases, an isotype control antibody was used. In addition, mouse staining was controlled with tissues derived from a Sparcl1 complete knockout mouse, and in human studies, comparisons between healthy tissue and respective tumor tissue of the same organs were used. Moreover, the results were validated by western blot and qRT-PCR. In all instances, consistent results were obtained within organs of the same species, indicating that specific staining results were obtained. Therefore, the different methods employed can also be used without a major change in results in the future.

The striking difference in the SPARCL1 expression patterns for the liver and pancreas between mice and humans emphasizes that the possibility of transferring findings from mouse models to human patients is not possible in all cases. According to our results, care must be taken when analysing the impact of SPARCL1 specifically in the liver context. Moreover, our findings suggest that it is highly recommended to perform systematic expression analyses in humans and mice before performing mechanistic mouse studies. Similar differences in the expression pattern of a protein in the human and murine organisms have been reported in a few previous studies, for example, for LIM-only proteins [31].

Finally, previous studies showed that hSPARCL1 expression in endothelial cells induced an inhibition of EC proliferation, migration and sprouting and consequently reduced angiogenesis. Human SPARCL1 has been found to be endogenously expressed in the ECs of quiescent vessels as well as in mural cells and stabilizing mature vessels, in turn leading to homeostasis [5]. Here, the functions of SPARCL1-secreting ECs in particular might be one reason for the downregulation of hSPARCL1 expression in neoplastic tissues in the course of tumor escape, which we found in accordance with previous studies. Future studies should elucidate whether the preferential loss of SPARCL1 in various malignomas is due to its function as a vascular-derived tumor suppressor favouring vessel homeostasis, as reported for colorectal carcinoma.

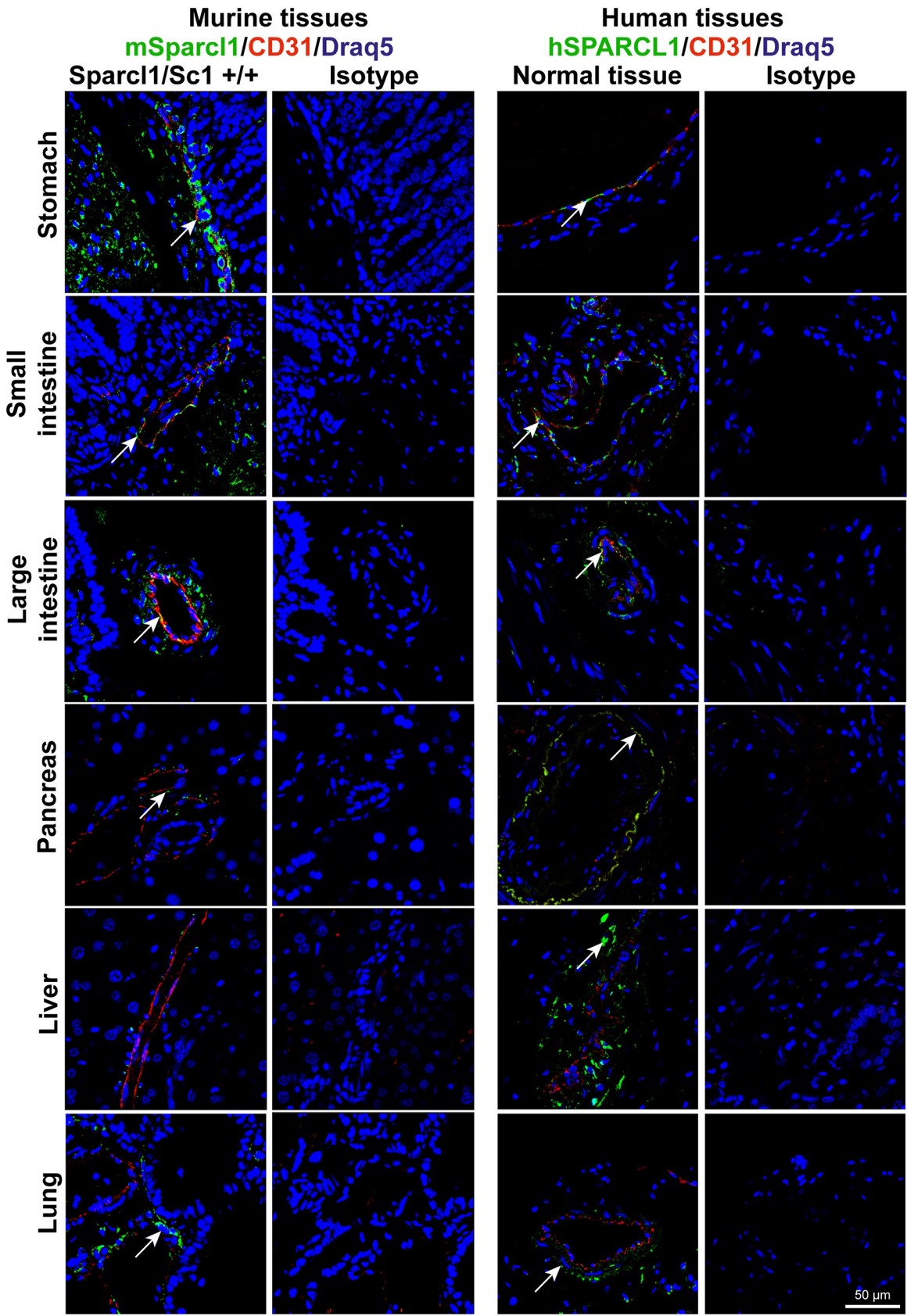

**Fig 4. SPARCL1 is differentially expressed between murine and human organs in a species- and cell-type-dependent manner.** Co-localization of human and murine (Sc1 +/+) SPARCL1 and CD31 was determined by immunofluorescent double staining (SPARCL1 green, arrows; CD31 red). Isotype antibody staining of consecutive sections was used as a negative control (isotype). All tissues were counterstained using DRAQ5 (blue). Scale bar = 50 μm.

**Table 1. Expression of SPARCL1 in different human and murine tissues.**

| Organ | Human SPARCL1 | | Murine Sparcl1 | | | |
| --- | --- | --- | --- | --- | --- | --- |
| | IHC | Cellular location | IHC | WB | RNA | Cellular location |
| Oesophagus | ++ | EC, MC, epithelium, SMC | + | + | + | EC, MC, SMC |
| Stomach | +++ | EC, MC, SMC | ++ | +++ | ++ | MC, SMC |
| Small intestine | ++ | EC, MC, SMC | + | - | + | EC, SMC |
| Large intestine | +++ | EC, MC, SMC | ++ | +++ | ++ | EC, MC, SMC |
| Pancreas | +++ | EC, MC, islet cells | + | + | ++ | MC |
| Liver | ++ | MC, hepatocytes | - | - | - | - |
| Lung | +++ | EC, MC, around bronchial tubes, alveolar tissue | +++ | +++ | +++ | MC, around bronchial tubes, alveolar tissue |
| Heart | n.d. | | ++ | + | ++ | MC, muscle cells |
| Spleen | n.d. | | + | + | ++ | MC |
| Kidney | n.d. | | + | - | + | MC |
| Urinary bladder | n.d. | | ++ | ++ | ++ | MC, SMC |

IHC, immunohistochemistry; WB, western blot

EC, endothelial cells; MC, mural cells; SMC, smooth muscle cells

+++, strong expression; ++, intermediate expression; +, weak expression; -, no expression

n.d. not determined

**Table 2. Downregulation of human SPARCL1 in malignant carcinomas.**

| | Healthy tissue | | Malignant carcinoma | |
| --- | --- | --- | --- | --- |
| | IHC | Cellular location | IHC | Cellular location |
| Oesophagus | ++ | EC, MC, epithelium, SMC | ++ | EC, MC |
| Stomach | +++ | EC, MC, SMC | ++ | EC, MC |
| Small intestine | ++ | EC, MC, SMC | ++ | EC, MC |
| Large intestine | +++ | EC, MC, SMC | ++ | EC, MC, SMC |
| Pancreas | +++ | EC, MC, islet cells | ++ | EC, MC |
| Liver | ++ | MC, hepatocytes | ++ | MC, hepatocytes |
| Lung | +++ | EC, MC, around bronchial tubes, alveolar tissue | ++ | EC, MC, alveolar tissue |

IHC, immunohistochemistry

EC, endothelial cells; MC, mural cells; SMC, smooth muscle cells

+++, strong expression; ++, intermediate expression; +, weak expression

Moreover, it will be necessary to investigate whether the differential association with vessel cells between humans and mice has functional implications in both physiological and patho-physiological contexts.

## Supporting information

**S1 Fig. Murine Sparcl1 is expressed in several murine organs.** Murine Sparcl1 expression (Sc1 +/+, green, arrows) was determined by immunofluorescence in various organs of Sparcl1

wild-type mice (n = 3). Isotype antibody (Sc1 +/+ isotype) and knockout mouse (Sc1-/-, n = 3) staining were used as controls. All tissues were counterstained using DRAQ5 (blue). Scale bar = 100 μm.
(TIF)

**S2 Fig. Murine Sparcl1 is expressed in all organs examined.** Murine Sparcl1 expression (Sc1 +/+, brown, arrows) was determined by permanent IHC in various organs of Sparcl1 wild-type mice (n = 3). Isotype antibody- (Sc1 +/+ isotype) and knockout mice (Sc1 -/-, n = 3) staining were used as controls. Asterisks indicate nonspecific staining. Scale bar = 75 μm.
(TIF)

**S3 Fig. Human SPARCL1 expression is highly detectable in all human organs examined and downregulated in their respective neoplasms.** Human SPARCL1 expression (brown, arrows) was determined by permanent IHC in various healthy and neoplastic organs of human patients (healthy, neoplastic, n = 3 for all organs except oesophagus, small intestine and lung tumor with n = 2). Isotype antibody staining of consecutive sections was used as a negative control (isotype). Asterisks indicate nonspecific staining. Scale bar = 100 μm.
(TIF)

## Acknowledgments

We thank Christian Flierl, Katja Petter and Gabriele Förtsch (all Division of Molecular and Experimental Surgery) for excellent technical assistance. We thank Valerie S. Meniel (Cardiff, UK) for providing the Sparcl1/Sc1 knockout mice.

The present work was performed in (partial) fulfilment of the requirements for obtaining the degree "Dr. med." for Anika Klingler.

## Author Contributions

**Conceptualization:** Michael Stürzl, Elisabeth Naschberger.

**Data curation:** Anika Klingler, Elisabeth Naschberger.

**Formal analysis:** Anika Klingler, Michael Stürzl, Elisabeth Naschberger.

**Funding acquisition:** Michael Stürzl, Elisabeth Naschberger.

**Investigation:** Anika Klingler, Michael Stürzl, Elisabeth Naschberger.

**Methodology:** Anika Klingler, Daniela Regensburger, Clara Tenkerian, Elisabeth Naschberger.

**Project administration:** Michael Stürzl, Elisabeth Naschberger.

**Resources:** Daniela Regensburger, Clara Tenkerian, Arndt Hartmann, Michael Stürzl, Elisabeth Naschberger.

**Supervision:** Clara Tenkerian, Nathalie Britzen-Laurent, Michael Stürzl, Elisabeth Naschberger.

**Validation:** Elisabeth Naschberger.

**Visualization:** Elisabeth Naschberger.

**Writing – original draft:** Elisabeth Naschberger.

**Writing – review & editing:** Daniela Regensburger, Clara Tenkerian, Nathalie Britzen-Laurent, Michael Stürzl, Elisabeth Naschberger.

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
