## [Decision Letter · Decision Letter 0]

14 Apr 2020

PONE-D-20-06477

Species-, organ- and cell type-dependent expression of SPARCL1 in human and mouse tissues

PLOS ONE

Dear Dr. Naschberger,

Thank you for submitting your manuscript to PLOS ONE. After careful consideration, we feel that it has merit but does not fully meet PLOS ONE’s publication criteria as it currently stands. Therefore, we invite you to submit a revised version of the manuscript that addresses the points raised during the review process.

Reviewers' comments:

Reviewer #1: SPARCL1 is a matricellular protein with tumorigenic function and is frequently down-regulated in tumors. Previous studies have found that SPARCL1 is differentially expressed in liver between mice and human, which is probably associated with colorectal carcinoma and its metastasis to liver. However, whether and how SPARCL1 is expressed in all organs and in other species has not been investigated. In the present study, the authors systematically analyzed the expression of SPARCL1 in multiple organs with various methods and compared the expression of SPARCL1 between mouse and human. And the results shows that in human samples, SPARCL1 was detected in all organs with the strongest expression in stomach, large intestine and lung, which is consistent with the expression pattern in mice; but murine liver showed an absence of SPARCL1 expression that is different with human liver, indicating that SPARCL1 function may not be similar in all tissues but dependent on species. The experiments were well designed, and were carefully controlled by setting SPARCL1 complete knock-out mouse and using healthy tissue and respective tumor tissue of the same organs. The results were clear and cautiously interpreted and the conclusion seems to be convictive. Generally speaking, this is an interesting manuscript regarding the expression of SPARCL1. However, revisions are needed before it could be accepted for publication in PLOS ONE.

Major points:

1. The main purpose of the study was to widely compare the expression of between murine model and human tissue. Therefore the author should focus on this purpose. It is acceptable to compare the SPARCL1 expression between different organs, different cells and between normal and neoplastic tissues, and the results are significant and interesting. However, during writing of this manuscript, it should always focus on the purpose of comparing the expression of between murine model and human tissue. Therefore the writing needs revision, especially those in the abstract and the introduction section.

2. The paragraph two, discussion section seems redundant. It can be removed and the contents may be written in relative paragraphs, respectively.

Minor points:

1. It is noted that the manuscript should be carefully edited to avoid grammar mistakes, e.g. the first sentence in paragraph 1, result section.

2. The left panel in figure 1A needs notes for symbols. E.g. the black dots represent mRNA level of wild type mSPARCL1 while grey dots represent that of SPARCL1 knockout type, which needs to be denoted clearly in the chart and/or figure legends.

Reviewer #2: Klingler Anika et al show that SPARCL1 expresses differently among species, organ and cell types. The experiments are well designed and performed with different controls. Relying on this study, we should be careful when we want to transfer findings from mouse models for the human patients. And some drug screen dependent on mice may not be applied into human diseases. It should be accepted for this journal and I have some concerns:

The major one is:

For human SPARCL1, a western blot is needed for its expression in each tissue and the migrating bands should be included in the WB. I am interested that the migrating bands may be functional.

And some minor concerns:

1) In Fig1B, migrating bands are different between Colon and Coecum, the authors should give an explanation for this.

2) Similar as the above one, the authors show that human SPARCL1 is downregulated in malignant diseases, a WB is needed to test whether there is difference in the migrating bands.

3) SPARCL1 in mouse should be written as Sparcl1, with only the first letter capitalized.

We would appreciate receiving your revised manuscript by May 29 2020 11:59PM. To enhance the reproducibility of your results, we recommend that if applicable you deposit your laboratory protocols in protocols.io, where a protocol can be assigned its own identifier (DOI) such that it can be cited independently in the future. For instructions see: http://journals.plos.org/plosone/s/submission-guidelines#loc-laboratory-protocols

We look forward to receiving your revised manuscript.

Kind regards,

Yuqin Yao

Academic Editor

PLOS ONE

Journal Requirements:

2. At this time, we request that you  please report additional details in your Methods section regarding animal care: 1) Please provide details of animal welfare (e.g., shelter, food, water, environmental enrichment) 2) please describe any steps taken to minimize animal suffering and distress, such as by administering anesthetics or analgesics, 3) please include the method of sacrifice and 4) Please describe the source of the animals used in this study. Thank you for your attention to these requests.

4. We note you have included a table to which you do not refer in the text of your manuscript. Please ensure that you refer to Table 2 in your text; if accepted, production will need this reference to link the reader to the Table.

---

## [Author Response · Author response to Decision Letter 0]

22 Apr 2020

Point-by-point response for the reviewers

Klingler et al, PLOS ONE

“Species-, organ- and cell type-dependent expression of SPARCL1 in human and mouse tissues”

#PONE-D-20-06477

We thank both reviewers for their positive evaluation of our manuscript. In addition, their suggestions were very helpful. All points were answered or addressed in full and have been incorporated into the revised version of the manuscript. The alterations introduced in the revised manuscript are shaded in gray.

REVIEWER 1:

Major points:

1. The main purpose of the study was to widely compare the expression of between murine model and human tissue. Therefore, the author should focus on this purpose. It is acceptable to compare the SPARCL1 expression between different organs, different cells and between normal and neoplastic tissues, and the results are significant and interesting. However, during writing of this manuscript, it should always focus on the purpose of comparing the expression of between murine model and human tissue. Therefore, the writing needs revision, especially those in the abstract and the introduction section.

The abstract and introduction has been revised according to the reviewer´s suggestions and it was tried to put a more stringent focus on the mentioned comparison between human and mouse. All alterations are shaded in grey.

2. The paragraph two, discussion section seems redundant. It can be removed and the contents may be written in relative paragraphs, respectively.

As suggested the paragraph two has been removed and the content was added in a more compact form as a single sentence at the beginning of the discussion section. 

Minor points:

1. It is noted that the manuscript should be carefully edited to avoid grammar mistakes, e.g. the first sentence in paragraph 1, result section.

The sentence is corrected now. Moreover, the whole manuscript was sent to “American Journal Experts” for professional revision of the language. The certificate for language revision is attached now for the reviewer´s attention.

2. The left panel in figure 1A needs notes for symbols. E.g. the black dots represent mRNA level of wild type mSPARCL1 while grey dots represent that of SPARCL1 knockout type, which needs to be denoted clearly in the chart and/or figure legends.

The respective notes for the symbols are added now in the corresponding legend of Figure 1A. 

REVIEWER 2:

The major one is:

For human SPARCL1, a western blot is needed for its expression in each tissue and the migrating bands should be included in the WB. I am interested that the migrating bands may be functional.

We fully agree with the suggestion of the reviewer and would highly appreciate to perform this type of analysis. We have tried already before submission of the manuscript to get the required material for western blot. However, our pathology cannot provide us with cryomaterial of the required human organs but instead only with FFPE-blocks. From the latter material western blot analysis cannot be performed at sufficient quality, unfortunately. Therefore, we ask the reviewer kindly to refrain from this experiment. 

And some minor concerns:

1) In Fig1B, migrating bands are different between Colon and Coecum, the authors should give an explanation for this.

We have revisited the original blots following the reviewer´s suggestion. The colon and coecum lysates were running on two different gels as also indicated by the separated boxes in the figure. We have referenced the SPARCL1-specific bands to our protein marker and aligned accordingly. However, the protein standard does not produce sharp bands in the low kDa range. Therefore, we cannot say with a high accuracy whether the small difference in size is a real difference or is only related to small variations produced by standard alignment. Accordingly, we did not state in the results section that there is a real difference in size.

2) Similar as the above one, the authors show that human SPARCL1 is downregulated in malignant diseases, a WB is needed to test whether there is difference in the migrating bands.

As stated above, we unfortunately do not have access to human material compatible with western blot. Therefore, we ask the reviewer kindly to refrain from this experiment.

3) SPARCL1 in mouse should be written as Sparcl1, with only the first letter capitalized.

The whole manuscript and the figures were revised accordingly. Of note, in order to be consistent with the mouse nomenclature also murine SPARC (osteonectin) was now abbreviated as mSparc with small letters after the first letter capitalized. All alterations are shaded in grey.

---

## [Decision Letter · Decision Letter 1]

6 May 2020

Species-, organ- and cell-type-dependent expression of SPARCL1 in human and mouse tissues

PONE-D-20-06477R1

Dear Dr. Naschberger,

We are pleased to inform you that your manuscript has been judged scientifically suitable for publication and will be formally accepted for publication once it complies with all outstanding technical requirements.

With kind regards,

Yuqin Yao

Academic Editor

PLOS ONE

Reviewers' comments:

Reviewer #1: The authors have addressed all my concerns and revised mistakes that I pointed out. I have no further comments. I think the manuscript is acceptable for publication in PLOS ONE.

Reviewer #2: All of my concerns are addressed or at least have explanations. For human SPARCL1, I understand that human samples are not easy to get for western blot analysis and I think conclusions in the paper are still solid, although western blot analyses make it better.

---

## [Editor Report · Acceptance letter]

8 May 2020

PONE-D-20-06477R1 

Species-, organ- and cell-type-dependent expression of SPARCL1 in human and mouse tissues 

Dear Dr. Naschberger:

I am pleased to inform you that your manuscript has been deemed suitable for publication in PLOS ONE. Congratulations! Your manuscript is now with our production department. 

With kind regards,

on behalf of

Dr. Yuqin Yao 

Academic Editor

PLOS ONE